# Marcus inverted region of charge transfer from low-dimensional semiconductor materials

Junhui Wang [1,4], Tao Ding [1,4], Kaimin Gao[1,2], Lifeng Wang[1,2], Panwang Zhou [3] & Kaifeng Wu [1,2✉]

A key process underlying the application of low-dimensional, quantum-confined semiconductors in energy conversion is charge transfer from these materials, which, however, has not been fully understood yet. Extensive studies of charge transfer from colloidal quantum dots reported rates increasing monotonically with driving forces, never displaying an inverted region predicted by the Marcus theory. The inverted region is likely bypassed by an Auger-like process whereby the excessive driving force is used to excite another Coulomb-coupled charge. Herein, instead of measuring charge transfer from excitonic states (coupled electron-hole pairs), we build a unique model system using zero-dimensional quantum dots or two-dimensional nanoplatelets and surface-adsorbed molecules that allows for measuring charge transfer from transiently-populated, single-charge states. The Marcus inverted region is clearly revealed in these systems. Thus, charge transfer from excitonic and single-charge states follows the Auger-assisted and conventional Marcus charge transfer models, respectively. This knowledge should enable rational design of energetics for efficient charge extraction from low-dimensional semiconductor materials as well as suppression of the associated energy-wasting charge recombination.

[1] State Key Laboratory of Molecular Reaction Dynamics and Dynamics Research Center for Energy and Environmental Materials, Dalian Institute of Chemical Physics, Chinese Academy of Sciences, 116023 Dalian, Liaoning, China. [2] University of the Chinese Academy of Sciences, 100049 Beijing, China. [3] Institute of Molecular Sciences and Engineering, Institute of Frontier and Interdisciplinary Science, Shandong University, 266235 Qingdao, Shandong, China. [4] These authors contributed equally: Junhui Wang, Tao Ding. ✉email: kwu@dicp.ac.cn

Low-dimensional quantum-confined materials, such as quantum dots, carbon nanotubes, graphenes, and more recently introduced monolayer transition metal dichalcogenides, have attracted tremendous attention for optoelectronic devices and solar energy conversion, mostly motivated by their strong light-matter interaction and size-tunable optical spectra and band edge energetics[1–4]. A key fundamental process underlying these applications is charge transfer from or into low-dimensional materials[5–7], which has been intensively studied in the literature[7–13].

The theoretical framework for charge transfer was outlined by Marcus et al. in the 1950s[14], which has been applied to describe charge transfer in various ionic, molecular and biological systems[15]. According to the Marcus theory, the nonadiabatic charge transfer rate ($k_{CT}$) in a donor-acceptor system at the high-temperature limit can be described by:

$$k_{CT} = \frac{2\pi}{\hbar} |V|^2 \frac{1}{\sqrt{4\pi\lambda k_B T}} \exp\left[-\frac{(\lambda + \Delta G)^2}{4\lambda k_B T}\right], \qquad (1)$$

where $\hbar$ is the reduced Plank constant, $|V|^2$ the donor-acceptor coupling strength, $\lambda$ the reorganization energy for the charge transfer reaction, $k_B$ the Boltzmann constant, $T$ the temperature, and $\Delta G$ the free energy difference between the product and reactant states ($-\Delta G$ is often called the driving force). The most important predication by the Marcus theory is a so-called inverted region within which $k_{CT}$ decreases with increasing $-\Delta G$ when $-\Delta G$ exceeds $\lambda$ (Fig. 1a; red solid curve). This counterintuitive predication had been under debate for decades until its experimental validation in the 1980s by Miller et al. using a series of donor-acceptor molecules with well-controlled energetics[16–18]. More recent studies even directly observed the inverted region in solid-state molecular junction or transistor devices and demonstrated the enhancement of device performance in the inverted region[19–21]. Relatedly, an inverted region was also reported for concerted proton-electron transfer (CPET) reactions, which is a long-sought goal in this field[22]. Thus, observing and understanding Marcus-inverted charge transfer phenomena are still at the forefront of physical sciences.

Although several reports proposed charge transfer for low-dimensional materials in the inverted region based on energetics analysis[23,24], direct measurements of charge transfer from colloidal quantum dots (QDs) using time-resolved spectroscopy revealed monotonic increase of charge transfer rate with increasing driving force and thus the lack of an inverted region

(Fig. 1a; green dashed curve)[25–27]. In order to rationalize this observation, Lian et al. proposed an Auger-assisted mechanism, whereby the driving force can be used to excite the charge of the opposite sign inside the band, effectively circumventing the inverted region (Fig. 1b, left)[26,28]. This effect is particularly strong in low-dimensional systems because the quantum and dielectric confinement effects strongly enhances Coulomb coupling between charges. A related example is that a hot electron in a QD can rapidly relax to the band edge by transferring its excessive kinetic energy to a hole[29], overcoming the otherwise expected phonon bottleneck effect[30,31] (which is essentially an inverted region for nonradiative hot electron relaxation).

In principle, one should measure charge transfer in the absence of other Coulomb-coupled charge(s) to uncover the fundamental energetic dependence of charge transfer from low-dimensional materials. It is interesting to test in this case whether charge transfer falls in the Marcus normal and inverted regions when the driving force is smaller and larger, respectively, than the reorganization energy. Such a measurement of charge transfer from single-charge rather than excitonic states not only will deepen fundamental understanding of charge transfer from low-dimensional materials but also is essential for their practical applications. This point is illustrated in the scheme in Fig. 1b where QDs are implemented into a photocatalytic system containing both electron and hole acceptors. Photoexcitation of QDs triggers sequential electron and hole transfer to their acceptors, whereas concerted electron-hole transfer is very unlikely[8,32]. Therefore, while the first charge transfer is from an excitonic state (Fig. 1b, left), the second one is from a single-charge state (Fig. 1b, right). Understanding the energetic dependences in both steps is essential for rational system design, as a slow rate in either step would become a bottleneck for the overall efficiency of the system[6]. Moreover, an energy-wasting charge recombination (CR) process following the first charge transfer competes with the second charge transfer (Fig. 1b, right), and thus, understanding their energetic dependences would be useful for CR suppression.

While carrier-doping is a common tool to create single-charge states, it does not apply to donor-acceptor systems in photo-induced charge transfer studies because the doped charge will always end up in the acceptor in a steady state. Inspired by the dynamic engagement of single-charge states in photocatalytic systems mentioned above, herein we build model systems based on complexes comprising 0D QDs or 2D nanoplatelets (NPLs) and surface-adsorbed molecules that allow for measuring charge transfer from transiently populated, single-charge states of QDs

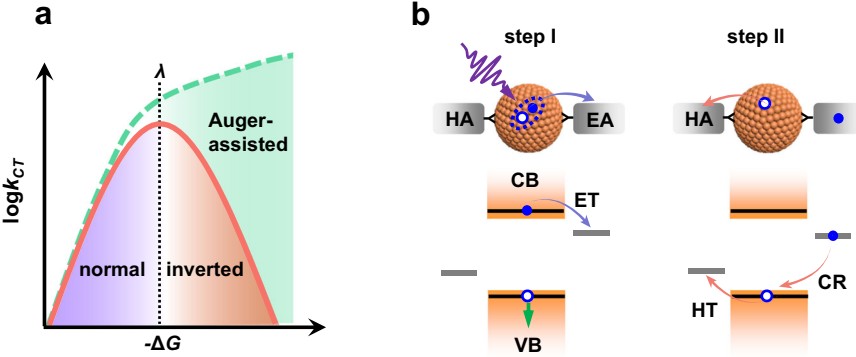

**Fig. 1 Charge transfer (CT) models. a** Marcus theory (red solid line) predicts a normal region (blue shading) and an inverted region (red shading) when the driving force ($-\Delta G$) is smaller and larger, respectively, than the reorganization energy ($\lambda$), whereas the Auger-assisted CT model (green dashed line) exhibits a monotonic increase of CT rate ($k_{CT}$) because the excessive driving force can be used to excite another Coulomb-couple charge. **b** For a photoexcited QD attached with electron (EA) and hole acceptors (HA), the first CT event (e.g., ET electron transfer) was proposed to occur via the Auger-assisted model (step I). In contrast, in step II, the second CT (e.g., HT hole transfer) obeys Marcus theory as it is not coupled any other charges, so does the energy-wasting charge recombination (CR) process. HT and CR are parallel, competing processes in step II.

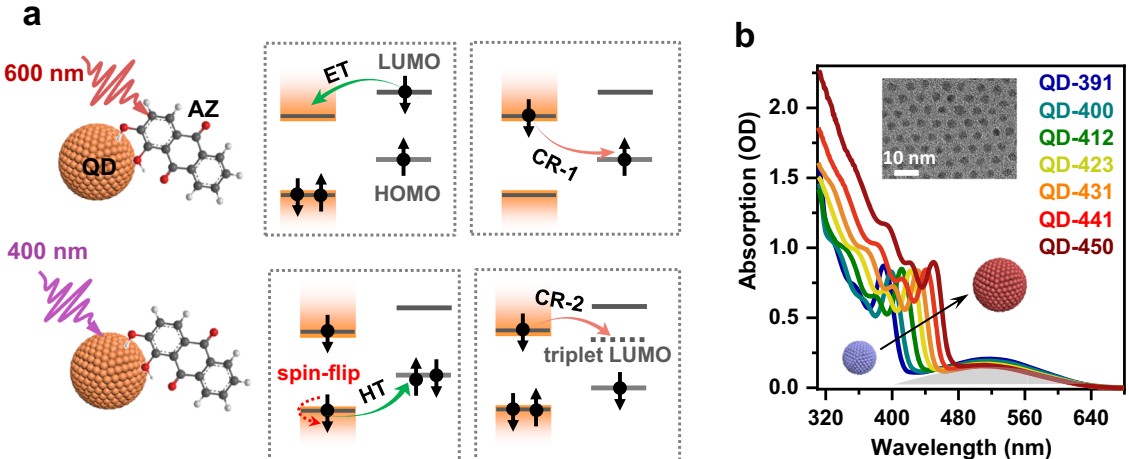

**Fig. 2 QD-alizarin complexes as a model system for charge transfer studies. a** Photoexcitation of QD-surface-bound alizarin (AZ) initiates electron transfer (ET) from the lowest unoccupied molecular orbital (LUMO) of molecules to the conduction band (CB) level(s) of QDs, which is followed by charge recombination (CR-1) from the lowest CB level of QDs to the highest occupied molecular orbital (HOMO) of molecules. Alternatively, exciting the QD triggers interfacial hole transfer (HT) from the valence band (VB) of the QD to the HOMO of AZs, which is also followed by another charge recombination process (CR-2) from the CB level of QDs to the triplet LUMO of AZs. **b** Absorption spectra of a series of CdS QD–AZ complexes dispersed in hexane with different QD sizes. The gray shading depicts the absorption band of surface-bound AZ. Inset is a representative transmission electron microscope (TEM) image of CdS QDs.

or NPLs, which allows us to reveal the hitherto unobserved Marcus-inverted region for these low-dimensional semiconductor materials.

## Results

**System design and characterization.** The principle of measuring charge transfer for the single-charge states of a QD is illustrated in Fig. 2a. For a QD-molecule hybrid system with a "type-II" (staggered) energy level alignment, photoexcitation of the surface-adsorbed molecule (Fig. 2a, top) leads to interfacial electron transfer (ET) from its lowest unoccupied molecular orbital (LUMO) into the conduction band (CB) of the QD, which is followed by charge recombination (CR-1) between the electron in the QD and the molecular cation. Alternatively, we can excite the QD (Fig. 2a, bottom), which triggers interfacial hole transfer (HT) from the valence band (VB) of the QD to the highest occupied molecular orbital (HOMO) of the molecule. This is also followed by a charge recombination process (CR-2). Among the four processes (ET, CR-1, HT, CR-2) discussed here, only HT is relevant to charge transfer from the excitonic state of the QD, whereas the others involve single-charge QD states that are transiently created by charge transfer.

Colloidal CdS QDs and Alizarin (AZ) molecules were chosen for this study because our previous studies showed that this system indeed allowed for selective excitation of either QDs or AZs to induce the charge transfer processes described in Fig. 2a[33,34]. In addition, this system exhibits a unique mechanism of "spin-controlled charge recombination pathways"[34]. Specifically, because the spin of the electron injected from photoexcited AZs to QDs has a longer lifetime than CR-1, for CR-1 electron migrates from the CB of QDs to the HOMO of AZs to regenerate ground-state complexes (Fig. 2a, top); see Supplementary Note 1 in the Supplementary Information (SI) for a brief discussion on the electron and hole spin lifetimes in CdS QDs. In contrast, spin-flip of the hole is rapid in the VB of II–VI group colloidal QDs due to a strong spin-orbit coupling (SOC). Scholes et al. measured this process for CdSe QDs using a cross-polarized transient grating method[35,36], and reported hole spin-flip time in the range of 0.1–1 ps when the QD diameter was smaller than 5 nm (the smaller the QDs, the faster the hole spin-flip is). Moreover, the

"triplet-like" dark excitons generated from hole spin-flip are situated at 10 s of meV lower than the "singlet-like" bright excitons[37], although it is noteworthy that spin is not good quantum number of QD states due to a strong SOC mentioned above. Combined with the higher statistic ratio of triplets than singlets, the exciton population should be dominated by "triplet-like" states. For these reasons, the QD exciton is qualitatively drawn as a spin-triplet in Fig. 2a (bottom). On the basis of this spin configuration, HT from QDs to AZs preferentially generates spin-triplet-like charge-separated states. Accordingly, CR-2 corresponds to electron transfer from the CB of QDs to the AZ cations to generate the spin-triplet excited states of AZs (Fig. 2a, bottom). We call the AZ level that the electrons migrate into the "triplet LUMO", which differs from the singlet LUMO by a singlet-triplet splitting energy of the molecule. Thus, CR-1 and CR-2 refer to electron transfer from the the CB of QDs to the HOMO and triplet LUMO, respectively, of AZs.

A straightforward means to study the dependence of the rates of CR1 and CR2 on driving forces is to tune the CB edge levels of QDs through their sizes using the quantum confinement effect. To this end, we synthesized CdS QDs with diameters ranging from 2.3 to 4.1 nm using a literature method[38]; synthetic procedures are provided in Methods. The lowest energy excitonic absorption peaks of CdS QDs were tuned from 391 to 450 nm via the quantum confinement effect, which are used to label the QD samples (e.g., QD-450). The UV-vis absorption spectra of these QD samples are provided in Supplementary Fig. 1 and their transmission electron microscope (TEM) images in Supplementary Fig. 2.

QD–AZ complexes were prepared by grafting AZ molecules onto QD surfaces using a simple ligand exchange procedure performed in hexane solution (Methods). The absorption spectra of QD–AZ complexes dispersed in hexane display a broadband absorption within 400–650 nm that can be attributed to QD-surface-adsorbed AZs in addition to the size-dependent excitonic bands of QDs (Fig. 2b). Note that the absorption spectrum of AZs adsorbed on QD surfaces is radically different from that of free AZs (Supplementary Fig. 3), due to a strong modification of their electronic structure through chelating with metal cations, which has been detailed in previous studies[33,34,39–41].

The energy levels of CdS QDs and surface-adsorbed AZs were analyzed on the basis of previous electrochemical studies on related samples[34,42]. Using these energy levels we can calculate the size-dependent driving forces for ET, CR1, and CR2. Note that in the calculation of energy levels and driving forces, we did not simply use the single-particle energy levels but rather accounted for the various forms of Coulomb energies (e.g., charging and binding energies) involved in charge transfer, as also illustrated in related prior studies[25,26,43,44]. The details are provided in Supplementary Note 2 and Supplementary Table 1. According to the calculation, the driving forces of CR1 and CR2 range from ~1.5 to 1.9 eV and 0.2 to 0.6 eV, respectively. The uncertainties of these numbers are at least 0.1 eV due to the errors from electrochemical measurements and from QD size distributions. When calculating the driving forces of ET from photoexcited AZs to QDs, we find that, for large-size QDs such as QD-450, ET into multiple energy levels in the CB is allowed. In this case, the observed ET rate should be the sum of these allowed kinetic pathways. Due to this complication, a comparison of the size-dependent ET rates is not meaningful here. The following part will thus focus on the size dependence of CR1 and CR2 rates.

**Measuring charge transfer.** Photoinduced charge transfer dynamics in the QD–AZ complexes were interrogated using broadband pump-probe transient absorption (TA) spectroscopy; see Methods. All the experiments were performed in the small-signal regime such that the TA kinetics were almost pump-fluence-independent in this regime. Figure 3a shows the TA spectra measured for the free CdS QD-450 sample (the largest size QD) at selected pump-probe delays following a 400 nm pump. The TA spectra are dominated by an exciton bleach (XB) feature of at ~450 nm that is rapidly generated (within 1 ps) and is then long-lived (decaying by only ~28.5% within 8 ns; see also Supplementary Fig. 4). Previous studies have established that the XB of CdS QDs is mainly induced by the state-filling effect of the CB edge electrons[8,45]. Therefore, the CB edge electrons in our QDs are almost free from fast trapping processes. This point is important as it guarantees that the dynamics of CR-1 and CR-2 to be studied below are not complicated by electron trapping processes (e.g., not trap-mediated charge transfer[46]).

The QD-450-AZ sample was excited at 600 nm to ensure selective excitation of surface-anchored AZ molecules. The resultant TA spectra in Fig. 3b exhibit a progressive formation of the XB feature of QDs at 450 nm within ~200 ps and then the decay of this feature within a few ns. This observation is consistent with ET from the LUMO of photoexcited AZs into the CB of QDs followed by CR-1 from QDs to the HOMO of AZs (Fig. 2a, top)[33,34,40]. These kinetics are also manifested on the TA spectral features of AZs (Fig. 3b inset). A brief discussion on these features is provided in Supplementary Fig. 5, and details can be found in our previous studies[33,34]. Particularly noteworthy is that the features of AZs largely recover within 8 ns, confirming that CR-1 mainly regenerates ground-state QD–AZ complexes.

The same sample is then subjected to a 400 nm pump, which selectively excites QDs (according to absorption spectra in Fig. 2b) and the resultant TA spectra are shown in Fig. 3c. In this case, HT from the VB of QDs to the HOMO of AZs causes the formation of AZ cation signals within ~2 ps, which subsequently evolves to the long-lived AZ triplet signals within ~1 ns (Fig. 3d and Supplementary Fig. 5)[34]. The latter is accompanied by the decay of the XB of QDs, consistent with

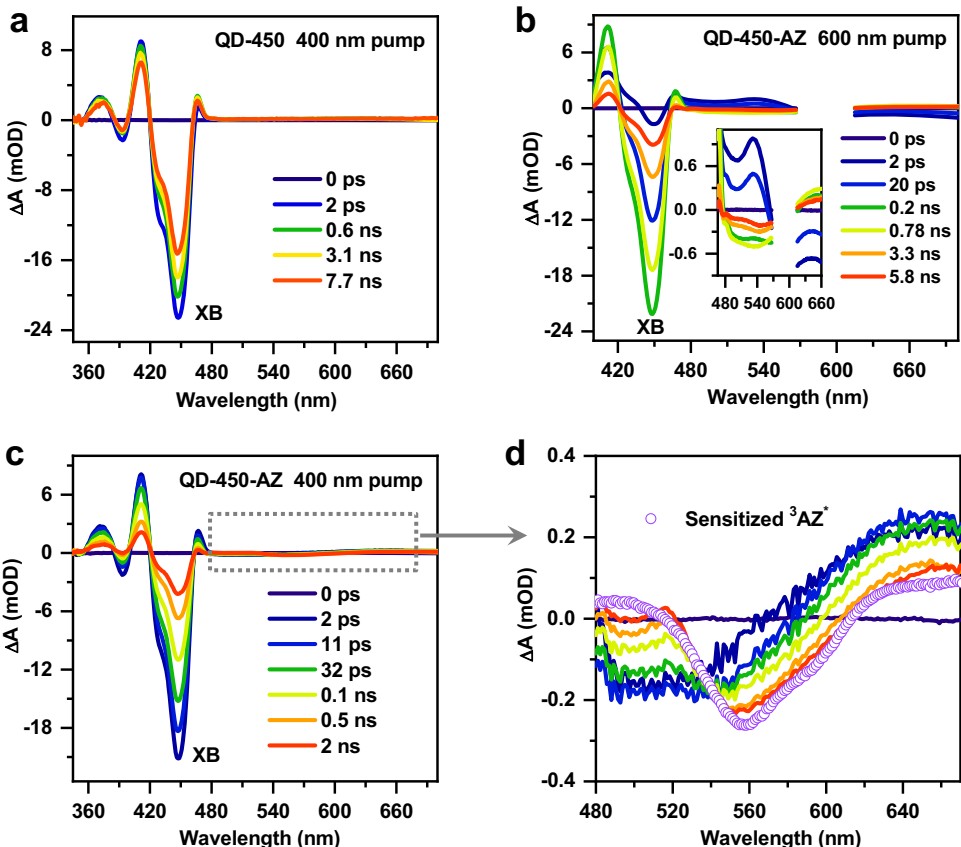

**Fig. 3 TA spectra in QD–AZ complexes. a** TA spectra of free CdS QD-450 nm following 400 nm excitation. **b, c** TA spectra of QD-450-AZ complexes following **b** 600 nm and **c** 400 nm excitations. Inset in **b** shows the enlarged view of AZ signals. **d** Enlarged view of AZ signals in **c**. The $^3AZ^*$ spectrum obtained by triplet sensitization (purple circles) is also shown for comparison.

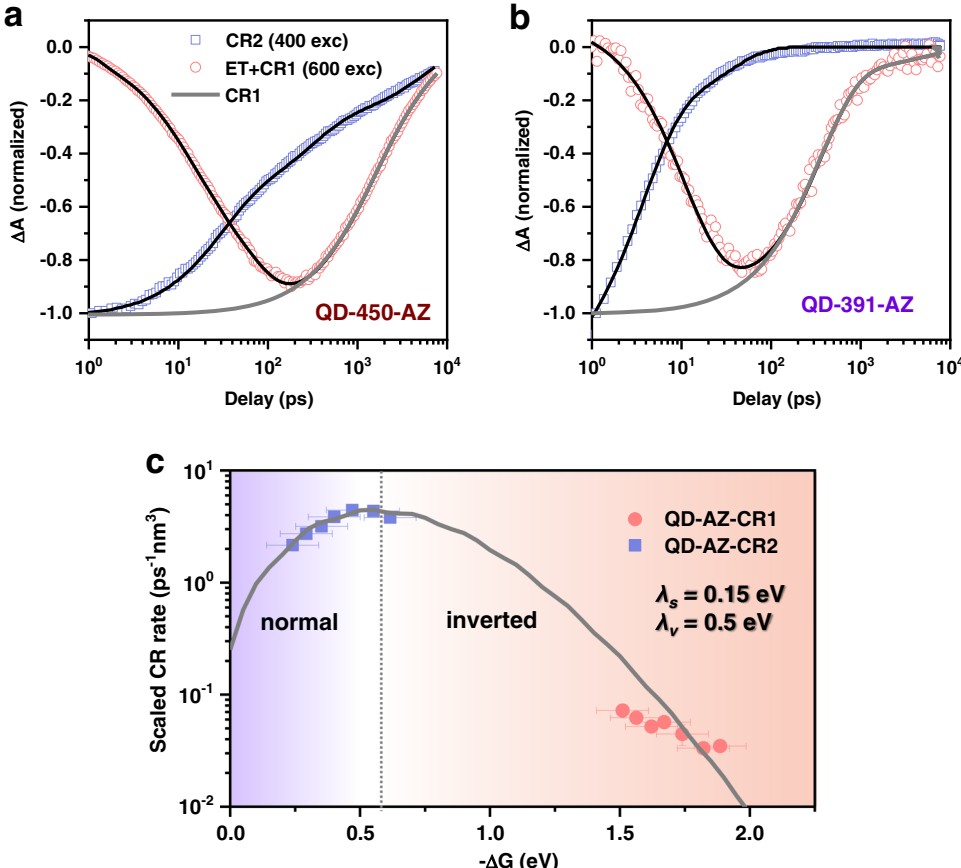

**Fig. 4 Observation of a Marcus-inverted region for CR rates in QD–AZ complexes. a, b** TA kinetics of the CdS QD exciton bleach feature for **a** QD-450-AZ and **b** QD-391-AZ complexes under 400 nm (blue squares) and 600 nm excitation (red circles). The black solid lines are multi-exponential fits to XB kinetics. The kinetics of CR1 pumped at 600 nm (gray solid lines) is obtained by deconvoluting the ET kinetics from XB kinetics. **c** Surface-electron-density scaled CR-1 (red circles) and CR-2 rates (blue squares) as a function of size-dependent driving forces ($-\Delta G$) in QD–AZ complexes. The gray solid line is a fit to Marcus theory (Eq. 2). Horizontal error bars stand for the accuracy of the electrochemical experiments for energy levels ($\pm 0.1$ eV), whereas vertical errors are the fitting errors of the CR rates.

the behaviour of CR-2 from the CB of QDs to the triplet LUMO of AZs (Fig. 2a, bottom)[34]. The distinction between CR-1 and CR-2 is controlled by the asymmetric spin-flip rates of the electron and hole in CdS QDs, as we briefly explained above and elaborated in our previous study[34].

The kinetics of the XB feature in the QD-450-AZ sample under both 400 and 600 nm excitations are compared in Fig. 4a. For QD–AZ complexes excited at AZs (600 nm), we can assign the growth and decay of the XB feature to ET and CR-1, respectively. In contrast, when selectively exciting QDs (400 nm), we observe the decay of the XB, which is assignable to CR-2. By fitting the kinetic traces to multi-exponential functions (Supplementary Note 3 in the SI), we obtain amplitude-averaged rate constants of $0.038 \pm 0.011$ ps$^{-1}$, $0.56 \pm 0.048$ ns$^{-1}$, and $0.016 \pm 0.0005$ ps$^{-1}$, respectively, for ET, CR-1, and CR-2 in the QD-450-AZ sample (fitting parameters in Supplementary Table 2).

Using the procedures described above, we measured the charge transfer kinetics in QD–AZ complexes with different QD sizes. Figure 4b shows the kinetics of ET, CR-1, and CR-2 in the QD-391-AZ sample (the smallest size QD); the plots for the rest intermediate size samples are provided in Supplementary Fig. 6. A universal observation for these samples is that CR-1 is orders of magnitudes slower than CR2, despite that the former has driving forces that are 1.27 eV larger than the latter. Considering that both CR1 and CR2 involve electron transfer from the single-electron states of QDs to the cations of AZs, except that the

destinations are the HOMO and triplet LUMO, respectively, our observation is a clear indication of a Marcus-inverted behavior of electron transfer from QDs. Thus, measuring charge transfer of QDs in the absence of Coulomb-coupled extra charge(s) indeed reveals intrinsic charge transfer properties not observed in previous studies of charge transfer from excitonic states of QDs[25–27]. We note that because CR1 and CR2 correspond to the coupling of the charge-separated states to the singlet ground state ($S_0$) and the triplet state ($T_1$) of surface-adsorbed AZs, respectively, their distinct coupling strengths might also result in different CR rates. In order to assess this possibility, we have performed density functional theory (DFT) calculations for the QD-surface-adsorbed AZ molecules (Supplementary Note 4). As plotted in Supplementary Fig. 7, the spatial electron density distributions of the $S_0$ and $T_1$ states are very similar, and both are mostly located in the vicinity of the chelated Cd atoms, a manifestation of the distinct electronic structure of QD-surface-adsorbed AZs. Thus, herein we can ignore the impact of distinct molecular orbitals on the rates of CR1 and CR2, and attribute their difference mainly to the driving forces.

**Size-dependent rates and Marcus-inverted region.** Although the comparison between CR1 and CR2 for each QD sample already confirms a Marcus-inverted behavior, a more enticing demonstration would be to compile all the size-dependent CR1 and CR2

rates to generate the whole Marcus curve. Note that, however, unlike comparing CR1 and CR2 for each sample, an accurate comparison of CR1 or CR2 between different QD samples requires us to exclude all possible size-dependent factors other than the size-dependent driving force.

The first one is associated with the average number of molecules anchored onto each QD. This number varies from sample to sample and roughly scales with the QD-surface area[43,47]. A usual practice is to divide the measured rate by the average molecule number such as to obtain an "intrinsic" rate[8,13,48]. This correction is not straightforward because there is a large uncertainty associated with determination of the molecule number. Importantly, our current experimental design nicely avoids this issue. Under our experimental conditions, regardless of how many AZs are attached on each QD, ET from photoexcited AZs to QDs or HT from photoexcited QDs to AZs results in only one AZ cation on each QD for the following CR1 or CR2 processes. Therefore, the measured CR1 and CR2 rates do not need any further correction, enabling a more meaningful comparison of size-dependent rates.

The other QD size-dependent factor is the electronic coupling term ($|V|^2$) in the Marcus equation. Specifically, according to previous studies on QDs, $|V|^2$ is proportional to the charge probability density on the QD surface, which can be expressed as wavefunction squared at the surface ($|\Psi_s|^2$)[26,49–51]. Smaller QDs should exhibit a higher $|\Psi_s|^2$ because a stronger confinement effect should results in more wavefunction "leakage" on to the surfaces[52]. Since both CR-1 and CR-2 involve the electron in the CB of QDs, we calculate $|\Psi_s|^2$ by treating the electron as a particle confined in a spherical well of a finite depth determined by the energy offset between the CB of wurtzite CdS and the LUMO of the ligand molecules using an effective mass approximation (EMA)[26]; see Supplementary Note 5 for details. This single-band EMA model was validated by comparing the computed size-dependent optical gaps to the experimental ones determined from the absorption spectra (Supplementary Fig. 8a). Their reasonable agreement indicates this simple model is sufficient to describe the energies and wavefunctions of the first quantized electron and hole states. As expected, the calculated $|\Psi_s|^2$ rapidly increases with decreasing QD sizes (Supplementary Tables 3). The size dependence of $|\Psi_s|^2$, and hence $|V|^2$, has to be factored out in order to examine the driving force dependence of the charge transfer rates. Because $|\Psi_s|^2$ depends upon the tunneling barrier (i.e., the offset between CdS CB and ligand LUMO), we examined the impact of the uncertainty of the barrier height on $|\Psi_s|^2$. As compared in Supplementary Fig. 8b, variation of the offset by 0.5 eV does not lead to noticeable changes to the size-dependent amplitudes of $|\Psi_s|^2$.

In Fig. 4c, we plot the rates of CR-1 and CR-2 scaled by $|\Psi_s|^2$ (i.e., $k_{CR}/|\Psi_s|^2$) as a function of the calculated driving forces ($-\Delta G$). Clearly, CR-1 is situated in the Marcus-inverted region, whereas CR-2 straddles the normal and inverted regions. The combination of CR1 and CR2 qualitatively depicts a curve predicted by the Marcus theory. However, in an effort to quantitatively fit the experimental data, we find that a modified version of Eq. 1 is needed; see Supplementary Fig. 9 for details. In the modified theory, charge transfer not only induces classical motions of the solvent molecules but also can place the product states in the vibrational excited states of the high-frequency quantum modes of the molecules[18,53,54]. We use the simplest form by assuming one averaged mode of energy $\hbar\omega$:

$$k_{CT} = \frac{2\pi}{\hbar}|V|^2 \frac{1}{\sqrt{4\pi\lambda_s k_B T}} \sum_{w=0}^{\infty} \frac{e^{-S}S^w}{w!} \exp\left[-\frac{(\lambda_s + w\hbar\omega + \Delta G)^2}{4\lambda_s k_B T}\right], \quad (2)$$

where $\lambda_s$ is the solvent reorganization energy, $S$ is the Huang-Rhys parameter that is related to the molecular vibrational part of

the reorganization energy ($\lambda_v$) via $S = \lambda_v/\hbar\omega$, and $w$ is the number of excited vibrational quanta in a specific pathway. This treatment separates the total reorganization ($\lambda$) into a classical solvent part ($\lambda_s$) and a quantum vibrational part ($\lambda_v$), and it reduces to Eq. 1 in the high-temperature limit[18,55]. Following previous studies on aromatic and quinone molecules[18], we adopt an averaged mode energy ($\hbar\omega$) at 1500 cm$^{-1}$ (or 0.186 eV). The data in Fig. 4c can be satisfactorily fitted by Eq. 2 using $\lambda_s = 0.15$ eV and $S = 2.7$ (i.e., $\lambda_v = 0.5$ eV). Both $\lambda_s$ and $\lambda_v$ are comparable with those derived in previous studies of molecular charge transfer in alkane solvents (0.15 and 0.45 eV, respectively)[18]. A recent study computed the reorganization energy of Rhodamine B molecule and found a value of 0.488 eV[56], which also agrees well with the value adopted here.

**Extension to 2D systems.** In order to generalize the observation to low-dimensional semiconductor materials, we also studied 2D nanoplatelets (NPLs). CdSe NPLs were chosen because of their well-established synthesis in the literature[57] and because of their suitable absorption spectra allowing for selective excitations of either NPLs or AZs in the TA experiments. We synthesized NPLs with four different thicknesses ranging from 1.2 to 2.1 nm (see Methods), corresponding to 3, 4, 5, and 6 monolayers of zinc-blende CdSe units[58]. Due to quantum confinement in the thickness direction, the lowest energy excitonic absorption peaks of CdSe NPLs were tuned from from 461 to 578 nm (Supplementary Fig. 10). Absorption spectra of NPL–AZ complexes dispersed in hexane are displayed in Fig. 5a.

Unlike that photoexcitation of CdS QDs resulted in ps hole transfer (HT) to AZs to form AZ cations, which was followed by CR-2 to generate AZ triplets, we find that photoexcitation of CdSe NPLs does not lead to observable AZ spectral features. It is likely that HT from CdSe NPLs to AZs is too slow to compete with electron-hole recombination in NPLs, although the detailed reasons remain unclear. Therefore, herein we focus on the CR-1 process following ET from photoexcited AZs to NPLs.

Figure 5b shows the TA spectra of the NPL-553-AZ sample pumped at 650 nm, which selectively excites AZs. Similar to the observations for CdS QDs, we observe gradual formation of the XB feature of CdSe NPLs at ~553 nm within ~100 ps, which then decays on the ns scale. The underlying picture is still ET from the LUMO of AZs into the CB of NPLs followed by CR-1 from NPLs to the HOMO of AZs. The kinetics probed at the XB is plotted in Fig. 5c and is fitted by a multi-exponential function, revealing amplitude-averaged rate constants of $0.055 \pm 0.0016$ ps$^{-1}$ and $0.0009 \pm 0.0001$ ps$^{-1}$ for ET and CR-1, respectively. Results for NPLs of other thicknesses are shown in Supplementary Fig. 11 and Supplementary Table 2.

Following the procedures described above for QD–AZ systems, we calculate the NPL-thickness-dependent driving forces of CR1, which are in the range of ~0.9–1.4 eV; see Supplementary Table 1 and Supplementary Note 2. We also rescale the measured CR1 rates by the thickness-dependent electronic coupling terms (Supplementary Table 3 and Supplementary Note 5). Figure 5d shows the plot of scaled CR1 rates as a function of $-\Delta G$, which is clearly situated in the Marcus-inverted region. The data can also be quantitatively reproduced by Eq. 2 using $\lambda_s$ of 0.15 eV and $\lambda_v$ of 0.5 eV as well. We note that the charge recombination rates described above are amplitude-averaged rates, which is a typical way to present none-single-exponential kinetics. Nonetheless, even if we use another way to present the rates (e.g., $1/\tau_{1/2}$, $\tau_{1/2}$ is the apparent half-lifetime read from the kinetic curve), the data can still be fitted to the Marcus theory (Supplementary Fig. 12).

Overall, our results on both 0D QDs and 2D NPLs unambiguously establish that the Marcus-inverted region is

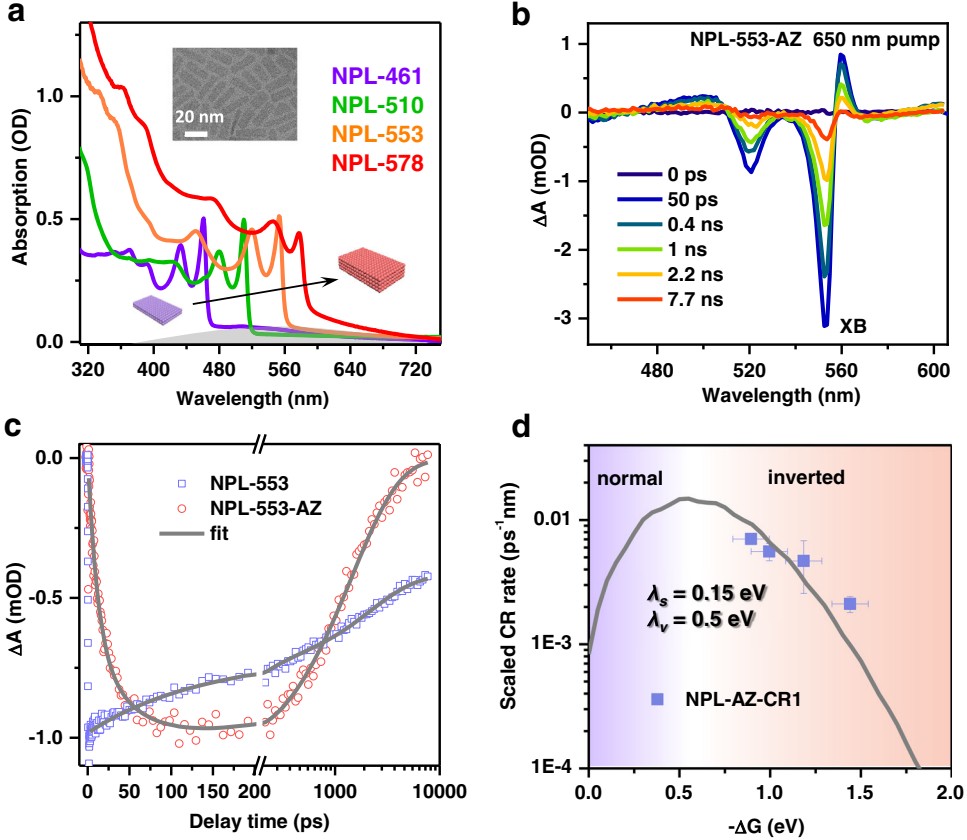

**Fig. 5 Observation of a Marcus-inverted region for CR rates in nanoplatelet-AZ complexes. a** Absorption spectra of a series of CdSe nanoplatelet (NPL)-AZ complexes dispersed in hexane with different NPL thicknesses. The gray shading depicts the absorption band of surface-bound AZ. Inset is a representative TEM image of CdSe NPLs. **b** TA spectra measured for the NPL-553-AZ sample following the excitation by a 650 nm pulse. **c** ET and CR-1 kinetics in the NPL-533-AZ sample probed at the exciton bleach (XB) feature (~553 nm; red circles). The kinetics of the XB of free NPL-553 under 400 nm excitation is also shown for comparison (blue squares). The gray solid lines are multi-exponential fits to the kinetics. **d** Surface-electron-density scaled CR1 rates (blue squares) as a function of thickness-dependent driving forces (−ΔG) in NPL–AZ complexes. The gray solid line is a fit to Marcus theory (Eq. 2). Horizontal error bars stand for the accuracy of the electrochemical experiments for energy levels (±0.1 eV), whereas vertical errors are the fitting errors of the CR rates.

observable for charge transfer from single-charge states of low-dimensional semiconductor materials. In order to further connect the data from QDs and NPLs, we scale the rates of NPLs by a constant, to account for the different ways of presenting electron densities for QDs and NPLs (Supplementary Note 5), and find that all the rates measured in this work can indeed be unified into the same Marcus curve (Fig. 6).

## Discussion

In conclusion, we demonstrate a method of studying charge transfer from the single-electron states of low-dimensional semiconductor nanocrystals to surface-anchored molecules in the absence of strongly Coulomb-coupled holes. Rather than directly measuring charge transfer from photoexcited excitonic states of these materials, we use photoexcitation to generate transient charge-separated states, for which the electrons are located in the conduction band of 0D quantum dots or 2D nanoplatelets whereas the holes are in the surface-adsorbed molecules, and then probe the electron dynamics in the ensuing charge recombination processes. Combined with our capability of tuning the electron transfer driving forces through the quantum confinement effect, this measurement allows us to probe the fundamental energetic dependence of electron transfer from quantum dots and nanoplatelets, and to unambiguously reveal a Marcus-inverted region that has never been observed before for these low-dimensional semiconductor materials. This result

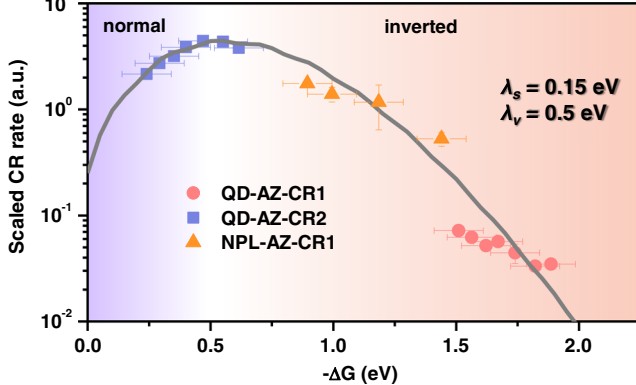

**Fig. 6 Combination of the results for CR1 (red circles) and CR2 (blue squares) in CdS QD–AZ complexes and for CR1 (orange triangles) in CdSe NPL–AZ complexes.** The gray solid line is a fit to Marcus theory (Eq. 2). Horizontal error bars stand for the accuracy of the electrochemical experiments for energy levels (±0.1 eV), whereas vertical errors are the fitting errors of the CR rates.

provides strong evidence for the Coulomb-coupling effects for charge transfer from excitonic states of these low-dimensional materials, in which the driving force of one charge transfer can be used to excite the other Coulomb-coupled charge (an Auger-like mechanism). The knowledge obtained here outlines a roadmap

for energetic design towards enhancing charge extraction and suppressing energy-wasting charge recombination using low-dimensional semiconductor materials, which is essential to their applications ranging from artificial photosynthesis to optoelectronic devices.

## Methods

**Sample preparations**. CdS QDs and CdSe NPLs were synthesized according literature methods[38,57,59], which are briefly described as follows. For a typical synthesis of CdS QDs, 0.077 g cadmium oxide (CdO), 15 mL 1-octadecene (ODE) and a varying amount of oleic acid (OA) (1–7.6 mL, more OA for large sizes) were loaded in a three-neck flask and were degassed under vacuum at 90 °C for 10 min. The solution was then heated to 270 °C in 25 min under argon flow until complete dissolution of CdO. After the temperature was stabilized, 3 mL sulfur (S) stock solution (0.1 M of S in ODE) was swiftly injected. The reaction was stopped after 30 s by injecting 7 mL room-temperature ODE and removing the heating mantle. The CdS QDs were centrifuged several times at ~8000 × g for 5 min with ethanol and were dispersed in hexane for use. For the synthesis of CdSe NPL-461, 0.0533 g cadmium acetate dehydrate (Cd(OAc)$_2$•2H$_2$O), 0.004 g selenium (Se), 0.0114 g stearic acid and 5 mL ODE were loaded in a three-neck flask and were degassed under vacuum for 10 min at room temperature. The solution was then heated to 180 °C under argon flow and the reaction was stopped after 30 min, and 1 mL OA was added when the reaction was cooled to room temperature. Finally, the NPLs were centrifuged at ~8000 × g for 10 min with ethanol and were dispersed in hexane for use. Details for the synthesis of CdSe NPLs of other thicknesses were provided in our recent work[60].

QD(NPL)-AZ complexes were prepared by adding alizarin (AZ) powders into QD or NPL solutions in hexane, followed by sonication for 30 min. The mixtures were filtered to obtain clear solutions containing QD(NPL)-AZ complexes in hexane; because the solubility of free AZs in hexane is very low, most of the molecules were bound to QD surfaces.

**TA experiment setups**. The femtosecond pump-probe TA measurements were performed using a regenerative amplified Ti:sapphire laser system (Coherent; 800 nm, 70 fs, 6 mJ/pulse, and 1 kHz repetition rate) as the laser source. Details were described in our previous reports[33,34]. Briefly, one part of the 800 nm output was used to pump a TOPAS Optical Parametric Amplifier to produce the wavelength-tunable pump beam. Another part with a weak intensity was focused onto a sapphire or CaF$_2$ window to generate the white light continuum as the probe beam. The delay between the pump and probe pulses was controlled by a motorized delay stage, and the pump powers were adjusted using neutral density filers. The samples were loaded in to 1 mm cuvettes and were vigorously stirred during the measurements.

## Data availability

The experiment data that support the findings of this study are available from the corresponding author upon reasonable request. In addition, Source data are provided with this paper.

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

## Acknowledgements

We gratefully acknowledge financial support from the National Natural Science Foundation of China (21773239, 51961165109, 21973091), the Strategic Pilot Science and Technology Project of Chinese Academy of Sciences (XDB17010100), the Ministry of Science and Technology of China (2018YFA0208703), and the Youth Innovation Promotion Association CAS (2021185).

## Author contributions

K.W. conceived the ideas and designed the project. J.W. performed the TA measurements. T.D. and L.W. synthesized the samples. K.G. and P.Z. made the calculations. J.W. and K.W. analyzed the data. K.W. wrote the paper with contributions from all authors.

## Competing interests

The authors declare no competing interests.
