## [Peer Review File · Nature Communications]

Marcus inverted region of charge transfer from low-dimensional semiconductor materialsREVIEWER COMMENTS

Reviewer #1 (Remarks to the Author):

Reviewer comments for manuscript NCOMMS-21-27484-T "Marcus Inverted Region of Charge Transfer from Low-Dimensional Semiconductor Materials"

This manuscript explores the dependence of quantum dot radius (and nanoplatelet thickness) on the rate of two distinct spin-controlled charge recombination processes between a photo-oxidized organic molecule (alizarin) and a photo-reduced quantum dot (or nanoplatelet). The two relevant spin-controlled charge separated states are prepared by either photoexciting the alizarin molecule and observing electron transfer (process 1) or photoexciting the quantum dot (nanoplatelet) and observing hole transfer (process 2). The charge separated state in process 1 is a singlet, while process 2 produces a triplet. This phenomenon was well-explored and nicely explained in the authors prior work (J. Am. Chem. Soc. 2020, 142, 10, 4723–4731). In both the prior work as well as this work, the authors describe the system and convincingly ascribe their spectral data to appropriate charge transfer processes. In both papers, process 1 is associated with a charge recombination (CR1) that is significantly slower than the charge recombination in process 2 (CR2). In both papers, this observation is proposed as evidence of the Marcus inverted region since there is a larger driving force for the slower charge recombination to the singlet ground state (CR1) as compared to the triplet ground state (CR2). The current paper extends this observation to a series of QD sizes (and nanoplatelet thicknesses) to construct a full driving force vs. CR rate plot that covers both the normal and inverted regions.

This manuscript has nicely presented data and analysis and tells a compelling story, but is not yet ready for publication in this reviewer's opinion until the following two major points are addressed. First, the main conclusion of the paper relies on an assumption that the coupling factor for charge recombination to a singlet ground state and a triplet state are the same. These are inherently different species, likely with distinct molecular orbitals and spatial extents. Some literature precedent or possibly molecular orbital computation is desired to justify this assumption. Second, the size-dependent trends are convolved with size-dependent coupling factors that are computed with a simplistic effective mass model that nevertheless has a larger effect on the scaled rates than the experimental data itself. Further description of the computational model and justification of its use are desired. For example, does it reproduce the size dependent band gaps appropriately? How sensitive is the surface wavefunction density to the approximated LUMO of oleic acid employed? Is the organic-inorganic interface modelled?

The following are additional minor points the authors may consider:

1. As mentioned above, this manuscript repeats a concerning number of experiments with the authors' previously published work. Further effort to distinguish the papers should be attempted. As an example, Figure 3d appears to be reproducing the same data as was previously published in J. Am. Chem. Soc. 2020, 142, 10, 4723–4731. Copyright permissions are recommended.
2. Figure 1b would be more clear if there was a step IIa and step IIb to distinguish HT from CR.
3. Include QD "HOMO" electrons in the top panel of Figure 2a for consistency sake.
4. "Monotonously" is used a few places, but I believe the authors mean to use "monotonically"

Reviewer #2 (Remarks to the Author):

The authors studied the dependence of charge recombination rate on the driving force in CdS quantum dot (or CdSe nanoplatelet)-alizarin (AZ) complexes. The main goal is to examine whether by removing electron-hole Coulomb interaction, electron transfer from nanocrystals follows the Marcus' behavior and the inverted regime can be observed. To answer this question, the author designed an elegant system that enables the study of charge transfer from the singly charge state in the nanocrystals by selective excitation of either the QD or the AZ molecule. The result clearly demonstrates the presence the Marcus inverted region in the driving force dependence of charge recombination rate. Although this result is fully expected, it is still very nice to see a careful demonstration.

The study is well designed and clearly presented in this manuscript. It can be published as it is in Nat. Comm.

Reviewer #3 (Remarks to the Author):

In this paper, the authors investigated the inverted region of Marcus theory for charge transfer between a quantum dot (QD) and an attached molecule. The inverted region (where larger driven force leads to smaller transition rate) is always a controversial topic for small molecular systems, although it is a common phenomenon in bulk defect system (it is well established, deeper the defect level, more difficult it is for the phonon induced charge trapping process). In particular, previous experiments only show a monotonic increase of a charge transfer transition rate as a function of the driving force. That has been explained by Auger process due to the existence of the opposite carrier in an optical induced carrier generation. The current authors carefully extract their time resolved exciton bleaching (XB) spectrum, and use different excitations and processes to induce different transitions. In particular, they used a spin flip phenomena when the QD is excited, to force the subsequent charge transfer to a higher level (CR2). They show, with reasonable credibility, that in the inverted region, when there is no opposite carrier in the quantum dot, the transition rate indeed follow the Marcus theory. I believe this is the first time such confirmation of the inverted region Marcus theory has been shown in such QD-molecule system. Because of that, I think this paper is potentially qualified for publication in Nat. Comm. Nevertheless, I believe there are several issues which should be addressed before the manuscript can be accepted.

(1) One of the shortcomings of the current manuscript is that, it is heavily dependent on their previous works. In some sense, this is just another continuation of their previous works, on similar systems. This however, makes the current manuscript difficult to read. I think some clarification will be necessary. For example, one critical process which made the CR2 transition possible is the spin flip of the QD hole state in the case of quantum dot excitation (400nm). I understand the spin-orbit coupling in the valence band made such spin flip possible, but the author should explain why it likes to flip at the first place. Is that to make the QD exciton from $S=0$ exciton to $S=1$ (triplet) exciton due to exchange splitting? How large is the exchange splitting for the given QD? Sometime it is only a few meV. Is that enough to ensure a dominant population of $S=1$ state at room temperature? Besides, quantitative, how fast is that spin splitting? I feel this should be stated clearly in the main text.

(2) The current study relies heavily on the comparison between CR1 and CR2 transitions. Fig.3d shows convincingly that the CR2 process indeed lead to a $3AZ^*$ state. However, their rate difference is not so big, perhaps by 100 times. On the other hand, CR1 is transferred to the HOMO state, and CR2 is transferred to the LUMO state (roughly), their wave functions and symmetries might be different. The coupling constant V in the Marcus theory not only depend on the wave function amplitude, but might also depend on the symmetry etc. Perhaps the authors should do a more careful theoretical investigation, e.g., using DFT calculation, instead of effective mass estimation, to discuss possible symmetry issues and coupling constants. It is not impossible different wave function couplings can cause a factor of 100 difference (consider both the wave function localization and symmetry forbidden issues). Overall, I feel the theoretical treatment in the manuscript is weak, and too simplistic. For example, the reorganization energy can perhaps also be estimated from DFT calculation for the AZ molecule.

(3) In the end, to match their data with the Marcus theory, Eq.(2), instead of Eq.(1) is used. If Eq.(1) is used, how good is the comparison? That can be shown in the SM. Besides, what is the justification of using Eq.(2)? Does the λ_s (re-organization energy) already taken into account these high frequency modes? Are there double counting?

(4) To make the manuscript easier to read, I suggest to complete Fig.2a, by adding the annotation of 400nm and 600nm excitation, QD exciton spin flip, the T state from LUMO etc. Right now, it is too simply and a bit confusing.

Reviewer comments in black, our responses in red and revisions in blue

REVIEWER COMMENTS

Reviewer #1 (Remarks to the Author):

This manuscript explores the dependence of quantum dot radius (and nanoplatelet thickness) on the rate of two distinct spin-controlled charge recombination processes between a photo-oxidized organic molecule (alizarin) and a photo-reduced quantum dot (or nanoplatelet). The two relevant spin-controlled charge separated states are prepared by either photoexciting the alizarin molecule and observing electron transfer (process 1) or photoexciting the quantum dot (nanoplatelet) and observing hole transfer (process 2). The charge separated state in process 1 is a singlet, while process 2 produces a triplet. This phenomenon was well-explored and nicely explained in the authors prior work (J. Am. Chem. Soc. 2020, 142, 10, 4723 - 4731). In both the prior work as well as this work, the authors describe the system and convincingly ascribe their spectral data to appropriate charge transfer processes. In both papers, process 1 is associated with a charge recombination (CR1) that is significantly slower than the charge recombination in process 2 (CR2). In both papers, this observation is proposed as evidence of the Marcus inverted region since there is a larger driving force for the slower charge recombination to the singlet ground state (CR1) as compared to the triplet ground state (CR2). The current paper extends this observation to a series of QD sizes (and nanoplatelet thicknesses) to construct a full driving force vs. CR rate plot that covers both the normal and inverted regions.

This manuscript has nicely presented data and analysis and tells a compelling story, but is not yet ready for publication in this reviewer's opinion until the following two major points are addressed.

Response: We thank the reviewer very much for his/her kind and constructive comments on our work. Below we address his/her remaining concerns point-by-point.

First, the main conclusion of the paper relies on an assumption that the coupling factor for charge recombination to a singlet ground state and a triplet state are the same. These are inherently different species, likely with distinct molecular orbitals and spatial extents. Some literature precedent or possibly molecular orbital computation is desired to justify this assumption.

Response: We thank the reviewer very much for this insightful comment. Indeed, CR1 and CR2 correspond to the coupling of the charge-separated states to the singlet ground state (S_0) and the triplet state (T_1) of surface-adsorbed AZs, respectively, for which the coupling factors might be different.

Per the reviewer's suggestion, we performed DFT and TD-DFT calculations on the CdS QD-surface-adsorbed AZ molecules, which were detailed in our prior paper mentioned by the reviewer ((J. Am. Chem. Soc. 2020, 142, 10, 4723). The accuracy of the calculation was established by comparing the computed transition energies to the

experimental ones. The computed transition energy of the S_1 state of free AZs in toluene is 2.84 eV, agreeing well with the experimental result (2.82 eV). The computed transition energy of the S_1 state of model compound of CdS-surface-bound AZs is 2.16 eV, which is also in reasonable agreement with the experimental result (2.05 eV).

Fig. R1a shows the optimized geometry of the model compound of CdS-surface-bound AZ. The two Cd atoms chelated by the AZ molecule are indicated. By setting the line along the two Cd atoms as the X-axis, the Y-axis is the normal of the QD-AZ interface along which electronic coupling takes place. Based on the DFT computed wavefunctions of S_0 and T_1 states, their corresponding electron densities were first computed and then integrated to reveal the electron density distributions along the Y-axis by employing the MultiWfn software using the following equation:

$$I_L(y) = \int_{-\infty}^{+\infty} \int_{-\infty}^{+\infty} p(x,y,z) dx dz$$

where $p(x,y,z)$ was the computed electron density. The results for S_0 and T_1 states are plotted in Fig. R1b for comparison, revealing almost the same electron density distributions for these two states. The same conclusion was found along other axes.

Therefore, for our current case, we can largely ignore the impact of molecular orbitals on the rates of CR1 and CR2, and attribute their difference mainly to the driving forces.

Figure R1. (a) The optimized geometry of the model compound of CdS-surface-bound AZ. The color codings of atoms are indicated except for H atom (white). Note the two hydroxyl groups attached to Cd are added for the purpose of balancing the charge. (b) Computed electron density distributions along the Y-axis (the normal of the QD-AZ interface) for S_0 (black) and T_1 (red) states by integrating along the other two axes.

Revision: On top of Figure 4, we add the following sentences

“We note that because CR1 and CR2 correspond to the coupling of the charge-separated states to the singlet ground state (S_0) and the triplet state (T_1) of surface-adsorbed AZs, respectively, their distinct coupling strengths might also result in different CR rates. In order to assess this possibility, we have performed density

functional theory (DFT) calculations for the QD-surface-adsorbed AZ molecules (Supplementary Note 4). As plotted in Supplementary Fig. 7, the spatial electron density distributions of the S_0 and T_1 states are very similar. Thus, herein we can ignore the impact of distinct molecular orbitals on the rates of CR1 and CR2, and attribute their difference mainly to the driving forces.”

In the SI, we add Figure R1 as the new Supplementary Fig.7 and the following content as the new Supplementary Note 4:

“Supplementary Note 4. Calculation of molecular orbitals

The density functional theory (DFT) and time-dependent DFT (TDDFT) calculations in present study were carried out with Gaussian 16 software. The ground state geometries of free alizarin and model compound of CdS-surface-bound alizarin were optimized using hybrid functional B3LYP, and then the frequency calculations at the same levels of theory were performed to confirm that each optimized structure was the real minimum. The vertically excited energies (VEEs) were computed using linear-response TD-DFT method based on the optimized ground state geometries. The TZVP basis set was adopted and the solvation effects were treated by the integral equation formalism version of the polarizable continuum model. The D3 version of Grimme's dispersion was included to account for the dispersion forces.

The accuracy of the calculation was established by comparing the computed transition energies to the experimental ones, as detailed in our previous study¹. Specifically, the computed transition energy of the S_1 state of free AZs in toluene is 2.84 eV, agreeing well with the experimental result (2.82 eV). The computed transition energy of the S_1 state of model compound of CdS-surface-bound AZs is 2.16 eV, which is also in reasonable agreement with the experimental result (2.05 eV).

Supplementary Fig. 7a shows the optimized geometry of the model compound of CdS-surface-bound AZ. The two Cd atoms chelated by the AZ molecule are indicated. By setting the line along the two Cd atoms as the X-axis, the Y-axis is the normal of the QD-AZ interface along which electronic coupling takes place. Based on the DFT computed wavefunctions of S_0 and T_1 states, their corresponding electron densities were first computed and then integrated to reveal the electron density distributions along the Y-axis by employing the MultiWfn software using the following equation:

$$I_L(y) = \int_{-\infty}^{+\infty} \int_{-\infty}^{+\infty} p(x, y, z) dx dz \quad (\text{S11}).$$

where $p(x,y,z)$ was the computed electron density. The results for S_0 and T_1 states are plotted in Supplementary Fig. 7b for comparison, revealing almost the same electron density distributions for these two states. The same conclusion was found along other axes.”

Second, the size-dependent trends are convolved with size-dependent coupling factors that are computed with a simplistic effective mass model that nevertheless has a larger effect on the scaled rates than the experimental data itself. Further description of the computational model and justification of its use are desired. For example, does it

reproduce the size dependent band gaps appropriately? How sensitive is the surface wavefunction density to the approximated LUMO of oleic acid employed? Is the organic-inorganic interface modelled?

Response: We thank the reviewer again for this insightful comment. Indeed, the single-band EMA model used in our study is a very simplified one. Obviously, it is too simple to fully describe the many quantized levels in the conduction and valence bands. Nevertheless, it is sufficient to capture the size-dependence of the carrier wavefunctions as well as energy levels of the *first* quantized electron and hole states. The following figure (Fig. R2a) shows the size-dependent optical gaps of CdS QDs calculated from the quantized levels and accounting for the electron-hole binding as a perturbation, which agrees reasonably well with our experimental data. Actually, the same agreement has been notified in the study of Lian group (Nano Lett. 2014, 14, 1263-1269), thus the simple model was also used therein to model size-dependent charge transfer from QDs.

As for the sensitivity of the wavefunction density to the LUMO level of OA ligands, we have performed two more sets of calculations by either down or upshifting the LUMO level by 0.25 eV (i.e., assuming a range of 0.5 eV for the offset between QD CB and ligands LUMO). The results are plotted in Fig. R2b. Within this range, the changes are essentially negligible.

Figure R2. (a) Comparison of the optical gaps of CdS QDs calculated from the first quantized electron and hole levels and accounting for the electron-hole binding as a perturbation (black solid line) with the experimental data (red circles). The errors represent the size distributions of the QDs. (b) Calculated surface electron densities ($|\Psi_s|^2$) with the OA LUMO at -0.75 eV (red squares), -1 eV (blue circles) and -1.25 eV (green triangles) vs. vacuum.

The reviewer asked a very insightful question about whether the inorganic/organic interface was modeled. Unfortunately, this is a very complicated problem well beyond our capability, and we think it actually remains a big challenge for the theoretic treatment of colloidal QDs (nobody really knows what the interface looks like now). Nevertheless, as demonstrated by the results in Fig. R2, the simple theory adopted here is sufficient to capture the physics required for our study.

Revision: In the paragraph where we describe the size-dependent wavefunctions, we add the following sentences

“This single-band EMA model was validated by comparing the computed size-dependent optical gaps to the experimental ones determined from the absorption spectra (Supplementary Fig. 8a). Their reasonable agreement indicates this simple model is sufficient to describe the energies and wavefunctions of the first quantized electron and hole states.... Because $|\Psi_s|^2$ depends upon the tunneling barrier (i.e., the offset between CdS CB and ligand LUMO), we examined the impact of the uncertainty of the barrier height on $|\Psi_s|^2$. As compared in Supplementary Fig. 8b, variation of the offset by 0.5 eV does not lead to noticeable changes to the size-dependent amplitudes of $|\Psi_s|^2$.”

In the SI, we add Figure R2 as the new Supplementary Fig. 8 and the following content to the new Supplementary Note 5:

“...In order to validate the accuracy of the simple model, we examined the size-dependent optical gaps of CdS QDs calculated from the first quantized electron and hole levels and accounting for the electron-hole binding as a perturbation, which agrees reasonably well with our experimental data (Supplementary Fig. 8a). Thus, this simple EMA model is likely sufficient to describe the energies and wavefunctions of the first quantized electron and hole states. Moreover, because $|\Psi_s|^2$ depends upon the tunneling barrier (i.e., the offset between CdS CB and ligand LUMO), we examined the impact of the uncertainty of the barrier height on $|\Psi_s|^2$. As shown in Supplementary Fig. 8b, changing the OA LUMO from -0.75 to -1.25 eV (vs. vacuum) does not lead to noticeable changes to the size-dependent amplitudes of $|\Psi_s|^2$. The reason is that the barrier is already high (~2.84 eV in the case of OA LUMO at -1 eV), and hence a variation of 0.5 eV does not strongly perturb the tunneling probability.”

The following are additional minor points the authors may consider:

1. As mentioned above, this manuscript repeats a concerning number of experiments with the authors' previously published work. Further effort to distinguish the papers should be attempted. As an example, Figure 3d appears to be reproducing the same data as was previously published in J. Am. Chem. Soc. 2020, 142, 10, 4723 - 4731. Copyright permissions are recommended.

Response: We thank the reviewer for this suggestion. Indeed, the phenomenon of spin-controlled charge recombination in CdS-AZ systems was reported in our prior paper. But we have carefully avoided reusing prior data in our current paper. The plot in Figure 3d is actually different from the one in our prior paper, in that the QD size are different (QD450 versus QD430). But we do agree with the reviewer that the similar plotting style might be an issue. We have now plotted Figure 3d in a different style to avoid this issue.

Revision: See the new Fig. 3 in the main text.

2. Figure 1b would be more clear if there was a step IIa and step IIb to distinguish HT from CR.

Response: We thank the reviewer for this suggestion. Our concern is that if we draw HT and CR separately in step IIa and step IIb, respectively, it might result in the confusion that these two take place sequentially, whereas in reality these two are parallel, competing processes.

Revision: We have modified the following caption of Fig. 1b to distinguish HT from CR:

“In contrast, in step II, the second CT (e.g., hole transfer, HT) obeys Marcus theory as it is not coupled any other charges, so does the energy-wasting charge recombination (CR) process. HT and CR are parallel, competing processes in step II.”

3. Include QD “HOMO” electrons in the top panel of Figure 2a for consistency sake.

Response: We thank the reviewer for this very good suggestion. We have made this change to the scheme.

Revision: See the new Fig. 2a in the main text.

4. “Monotonously” is used a few places, but I believe the authors mean to use “monotonically”

Response: We thank the reviewer for this comment. This word has now been corrected.

Reviewer #2 (Remarks to the Author):

The authors studied the dependence of charge recombination rate on the driving force in CdS quantum dot (or CdSe nanoplatelet)-alizarin (AZ) complexes. The main goal is to examine whether by removing electron-hole Coulomb interaction, electron transfer from nanocrystals follows the Marcus’ behavior and the inverted regime can be observed. To answer this question, the author designed an elegant system that enables the study of charge transfer from the singly charge state in the nanocrystals by selective excitation of either the QD or the AZ molecule. The result clearly demonstrates the presence the Marcus inverted region in the driving force dependence of charge recombination rate. Although this result is fully expected, it is still very nice to see a careful demonstration.

The study is well designed and clearly presented in this manuscript. It can be published as it is in Nat. Comm.

Response: We thank the reviewer very much for his/her very kind comments on our work and for recommend publication as is. Nevertheless, in order to resolve the issues raised by Reviewers #1 and #3, we have still made required changes to our paper.

Reviewer #3 (Remarks to the Author):

In this paper, the authors investigated the inverted region of Marcus theory for charge transfer between a quantum dot (QD) and an attached molecule. The inverted region (where larger driven force leads to smaller transition rate) is always a controversial topic for small molecular systems, although it is a common phenomenon in bulk defect system (it is well established, deeper the defect level, more difficult it is for the phonon induced charge trapping process). In particular, previous experiments only show a monotonic increase of a charge transfer transition rate as a function of the driving force. That has been explained by Auger process due to the existence of the opposite carrier in an optical induced carrier generation. The current authors carefully exact their time resolved exciton bleaching (XB) spectrum, and use different excitations and processes to induce different transitions. In particular, they used a spin flip phenomena when the QD is excited, to force the subsequent charge transfer to a higher level (CR2). They show, with reasonable credibility, that in the inverted region, when there is no opposite carrier in the quantum dot, the transition rate indeed follow the Marcus theory. I believe this is the first time such confirmation of the inverted region Marcus theory has been shown in such QD-molecule system. Because of that, I think this paper is potentially qualified for publication in Nat. Comm. Nevertheless, I believe there are several issues which should be addressed before the manuscript can be accepted.

Response: We thank the reviewer very much for these kind and constructive comments on our work. We address point-by-point the issues listed below.

(1) One of the shortcoming of the current manuscript is that, it is heavily dependent on their previous works. In some sense, this is just another continuation of their previous works, on similar systems. This however, makes the current manuscript difficult to read. I think some clarification will be necessary. For example, one critical process which made the CR2 transition possible is the spin flip of the QD hole state in the case of quantum dot excitation (400 nm). I understand the spin-orbit coupling in the valence band made such spin flip possible, but the author should explain why it likes to flip at the first place. Is that to make the QD exciton from S=0 exciton to S=1 (triplet) exciton due to exchange splitting? How large is the exchange splitting for the given QD? Sometime it is only a few meV. Is that enough to ensure a dominant population of S=1 state at room temperature? Besides, quantitative, how fast is that spin splitting? I feel this should be stated clearly in the main text.

Response: We thank the reviewer very much for this insightful comment. We agree with the reviewer that, in order to make the paper more accessible to readers, it is better to explain several key points regarding hole spin-flip in the CdS QDs.

The first point is the timescale for hole spin-flip. As pointed out by the reviewer, because of a strong SOC in the valence band of CdS QDs, the hole spin-flip is rapid,

but it is indeed better to specify how rapid it is. Hole spin-flip dynamics in CdSe QDs, sharing a similar electronic structure to CdS QDs studied here, were investigated in detail by Scholes et al. using a cross-polarized transient grating method (see, e.g., *Acc. Chem. Res.* 2009, 42, 1037-1046). According to their result, the hole spin-flip is in the range of 0.1 to 1 ps when the QD diameter is smaller than 5 nm; the smaller the QDs, the faster the hole spin-flip is. Therefore, for our CdS QDs with diameters ranging from 2.3 to 3.9 nm, it is likely that the hole spin-flip also occurs on a sub-ps timescale. This is the reason why we draw the spin-flipped hole in the first place in the scheme of Fig. 2a.

The second one is the splitting between singlet and triplet states. We note that because of strong SOC of the hole in II-VI QDs, it is actually not accurate to call the exciton states either a singlet or triplet because spin is no longer a good quantum number. Nevertheless, one can still qualitatively project a specific excitonic state to a spin singlet or triplet state. For example, it is easy to show that the $F = \pm 2$ (the projection of the electron-hole total angular momentum) dark states are pure spin-triplet states (see, e.g., *Adv. Funct. Mater.* 2008, 18, 1157-1172). The next higher-energy $F = \pm 1$ states are optically bright states and contains singlet features. We then need to examine the splitting between $F = \pm 2$ and ± 1 states. According to the latest work by Efros et al. taking in account both short and long-range exchange interaction (*Nano Lett.* 2018, 18, 4061-4068), this splitting is about 15 meV for CdSe QDs of 2-4 nm diameters. Again, considering the similar electronic structure of CdS and CdSe, we take this value as an approximate measure for our CdS QDs. If we simply assume a statistical ratio of 3:1 for triplet and singlet states, further considering the Boltzmann factor for an energy difference of 15 meV, we can derive the population ratio between triplet and singlet states is about 0.19:1. Thus, the dominant exciton species are indeed triplet states, from which triplet-like charge separated states are generated.

Revision: Per the reviewer's comments, we add the following content to the paragraph where we describe the spin-selective charge recombination process:

“In contrast, spin-flip of the hole is rapid in the VB of II-VI group colloidal QDs due to a strong spin-orbit coupling (SOC). Scholes et al. measured this process for CdSe QDs using a cross-polarized transient grating method,^{30,31} and reported hole spin-flip time in the range of 0.1 to 1 ps when the QD diameter was smaller than 5 nm (the smaller the QDs, the faster the hole spin-flip is). Moreover, the “triplet-like” dark excitons generated from hole spin-flip are situated at 10s of meV lower than the “singlet-like” bright excitons,³² although it is noteworthy that spin is not good quantum number of QD states due to a strong SOC mentioned above. Combined with the higher statistic ratio of triplets than singlets, the exciton population should be dominated by “triplet-like” states. For these reasons, the QD exciton is qualitatively drawn as a spin-triplet in Fig. 2a (bottom). On the basis of this spin configuration, HT from QDs to AZs preferentially generates spin-triplet-like charge separated states.”

(2) The current study relies heavily on the comparison between CR1 and CR2 transitions. Fig.3d shows convincingly that the CR2 process indeed lead to a 3AZ* state. However, their rate difference is not so big, perhaps by 100 times. On the other hand, CR1 is transferred to the HOMO state, and CR2 is transferred to the LUMO state (roughly), their wave functions and symmetries might be different. The coupling constant V in the Marcus theory not only depend on the wave function amplitude, but might also depend on the symmetry etc. Perhaps the authors should do a more careful theoretical investigation, e.g., using DFT calculation, instead of effective mass estimation, to discuss possible symmetry issues and coupling constants. It is not impossible different wave function couplings can cause a factor of 100 difference (consider both the wave function localization and symmetry forbidden issues). Overall, I feel the theoretical treatment in the manuscript is weak, and too simplistic. For example, the reorganization energy can perhaps also be estimated from DFT calculation for the AZ molecule.

Response: We thank the reviewer again for this very insightful comment. The same concern has been raised by Reviewer #1 in his/her first comment. We would like to refer the Reviewer to our responses above.

The reviewer also raised an interesting idea regarding the impact of orbital symmetry on wavefunction overlap. We agree that this is absolutely true. Nevertheless, the specific situation here allows us to simplify the treatment. To begin with, we are dealing with the $1S_e$ orbital of spherical QDs, for which the wavefunction is isotropic. Also, as shown in Fig R1, the electron density distributions for the S_0 and T_1 states of the QD-surface-adsorbed AZ molecule are almost the same. Overall, these conditions allow us to greatly simplify our treatment for QD-AZ electronic coupling.

The reviewer suggests calculating the reorganization energy of the molecule using DFT. But as we pointed in the paper, the molecular reorganization energy of 0.5 eV used here agrees well with the one (0.45 eV) determined for a series of small molecules (quinones, naphthalene, pyrene, phenanthrene...) in the classic work by Closs et al. (Science 1988, 240, 440-447). Moreover, in a recent study, the reorganization energy of Rhodamine B molecule was computed, reporting a value of 0.488 eV (J. Chem. Phys. 2019, 151, 074705), which is also consistent with 0.5 eV used here.

Revision: In addition to the revisions made according to Reviewer #1's first comment, we add the following sentence to address the comment related to the reorganization energy:

“A recent study computed the reorganization energy of Rhodamine B molecule and found a value of 0.488 eV,⁵¹ which also agrees well with the value adopted here.”

(3) In the end, to match their data with the Marcus theory, Eq.(2), instead of Eq.(1) is used. If Eq.(1) is used, how good is the comparison? That can be shown in the SM. Besides, what is the justification of using Eq.(2)? Does the λ_s (re-organization energy) already taken into account these high frequency modes? Are there double

counting?

Response: We thank the reviewer again for this comment. In Fig. R3, we show the fits of experimental results of Figure 4 using Eq. 1. No matter how we chose the total reorganization energy in the range of 0.3 to 1.1 eV, the fit does not work well. Basically, it cannot capture the slow decay of rate with increasing driving force in the range of ~ 0.6 to 2.0 eV (the inverted region). In contrast, Eq. 2 can capture this feature because it sums up the contributions of all the vibrationally-excited charge transfer channels; in the inverted region, these channels become more and more important as the driving force increases.

Figure R3. Fitting the experimental CR-1 (red circles) and CR-2 rates (blue squares) as a function of size-dependent driving forces ($-\Delta G$) in QD-AZ complexes. The colored solid lines are fits to Marcus theory in Eq. 1 with the total reorganization energy λ as the adjustable parameter.

We would like to note that the reorganization energies of λ_s and λ_v in Eq. 2 are not doubly-counted here; λ_s is the solvent reorganization energy (pure classic motion of solvent molecules), whereas λ_v is the reorganization of the AZ molecule accounting for the quantum nature of the vibration modes of the molecule. This point has been clearly explained in many review papers on the charge transfer theory (e.g., Science 1988, 240, 440-447).

Revision: We add the following sentence to briefly explain the reorganization energy terms in Eq. 2:

“This treatment separates the total reorganization (λ) into a classical solvent part (λ_s) and a quantum vibrational part (λ_v), and it reduces to Eq. 1 in the high temperature limit.^{18,50}”

In the SI, we add Fig. R3 as a new Supplementary Fig. 9.

(4) To make the manuscript easier to read, I suggest to complete Fig.2a, by adding the annotation of 400nm and 600nm excitation, QD exciton spin flip, the T state from LUMO etc. Right now, it is too simply and a bit confusing.

Response: We thank the reviewer very much for this suggestion. These notations have been added to Fig. 2a.

Revision: See the new Fig. 2a.

REVIEWERS' COMMENTS

Reviewer #1 (Remarks to the Author):

The authors have adequately responded to this reviewer's feedback on the earlier draft. Specifically, they demonstrate via computational work that the molecular orbitals associated with the triplet and singlet states are indistinguishable such that there will be no significant difference in the electronic coupling for the two charge recombination processes being investigated. Secondly, the validity of the effective mass model is tested by both comparing to size dependent band gaps and adjusting the energies of the ligands (Which may have some experimental uncertainty). In this reviewer's opinion, this work is well justified and nicely presented. It demonstrates the long sought after Marcus inverted region in systems involving quantum dots, and further justifies the Auger-assisted interpretation presented in prior charge transfer studies involving quantum dots and molecules. Specifically, the role an additional charge plays in eliminating the inverted region is confirmed by the current work which removes the additional charge (by focusing on charge recombination) and thereby recovers the inverted region. This work is of high quality and worthy of publication with no further modifications.

Reviewer #3 (Remarks to the Author):

I am mostly satisfied with the authors response to my questions and referee#1's questions. But I take issue on one point of their response, and I think this is a critical point, should be addressed more carefully before the manuscript can be accepted.

This is my original point (2), and also related to referee#1's first point: about the coupling between QD and AZ molecule wave function, especially the difference between the CR1 and CR2 transitions. The authors supposedly calculated the after transition S_0 state and T_1 state of the AZ molecule, and show that their electron charge densities are almost identical for these two states (Fig.S7). Frankly, I do not believe in this result. For one thing: S_0 and T_1 are consisted with different AZ molecular orbitals, one with AZ HOMO, one with AZ LUMO. How can they have exact same electron charge density? I feel there must be some mistakes. Note, these should not be the singlet and triplet QD exciton state before transition. Their electron charge densities can be the same if one only takes one electron QD state to construct the exciton state. But for AZ molecule, the S_0 and T_1 should have different single particle configurations, thus their electron charge density cannot be almost identical. Besides, it actually might be interesting just to plot the AZ HOMO and LUMO levels, do some simple dot-product with the QD electron wave function, to make sure their coupling are similar.

Reviewer #4 (Remarks to the Author):

I enjoyed to read the manuscript “Marcus Inverted Region of Charge Transfer from Low-Dimensional Semiconductor Materials” by Wang et al. The authors report CT in the inverted region between molecule—QD interfaces. I find the work interesting and the topic is timely as only recently Marcus Inverted region has been also found in solid-state devices (Nat. Nanotechnol. 13, 322–329 (2018) or Nat. Commun. 10, 2089 (2019)), in proton coupled charge transport (their ref 19), and now also in molecule—QD systems. At the very end of their paper the authors conclude that CT in the inverted Marcus region could be useful to improve the performance of devices. This point has been shown by Han et al. in Adv. Sci. 2021, 2100055 where the authors showed that the performance of a molecular diode was improved (30x) operating in the inverted Marcus region. I have seen the paper and the, in general, positive review reports: I also find the paper original and important, and support publication in Nat. Commun. as the other 3 reviewers also did. In my opinion, the authors have addressed all reviewers concerns adequately. In response to Reviewer 1, the authors include now detailed TD-DFT calculations. Although the QS is only represented by two Cd ions, the calc show, nevertheless, that the electron density distributions of S0 and T1 are very similar.

I do agree with the authors that is very challenging to understand the atomistic structure of molecule—QD interfaces (as the state in response to comment 2 by reviewer 1), I do believe the model structures can be studied to investigate, for instance, the energy-level alignment of the molecule—QD interface, or what the most favorable binding geometries are, which in turn are important to know to quantify the molecule—QD coupling strength. Such a detailed study is out of scope of the current paper and that is why a simplified approach suffice at this stage.

Reviewer 3, comment 1, questions whether low energy processes can be follow at room temperature because at RT thermal energy (26 meV) more than enough to washout any low energy effects. The authors argue that systems with low energy barriers still be studied at RT provide one can measure such processes fast enough which I agree with. The authors provide now a clear discussion to explain this.

Reviewer 3, comment 2, has been also (partially) addressed in response to Reviewer 1. I agree with the authors that to calc the reorganization energy for the full system is very difficult to do and would justify a publication on its own right. Instead, they compare their obtained reorganization energies to those obtained from other planar molecules which is fine. In general, quantification of the reorganization energy with ab initio theories in solid-state systems is challenging and certainly falls out of scope of this work.

Comment 3 by the same reviewer relates to comment 2 and has been, in my opinion, also sufficiently well addressed.

To summarize, I believe the paper is now suitable for publication in its current form, and I believe it will form the base for many future studies.

Reviewer comments in black, our responses in red and revisions in blue

REVIEWER COMMENTS

Reviewer #1 (Remarks to the Author):

The authors have adequately responded to this reviewer's feedback on the earlier draft. Specifically, they demonstrate via computational work that the molecular orbitals associated with the triplet and singlet states are indistinguishable such that there will be no significant difference in the electronic coupling for the two charge recombination processes being investigated. Secondly, the validity of the effective mass model is tested by both comparing to size dependent band gaps and adjusting the energies of the ligands (Which may have some experimental uncertainty).

In this reviewer's opinion, this work is well justified and nicely presented. It demonstrates the long sought after Marcus inverted region in systems involving quantum dots, and further justifies the Auger-assisted interpretation presented in prior charge transfer studies involving quantum dots and molecules. Specifically, the role an additional charge plays in eliminating the inverted region is confirmed by the current work which removes the additional charge (by focusing on charge recombination) and thereby recovers the inverted region. This work is of high quality and worthy of publication with no further modifications.

Response: We thank the reviewer very much for the kind comments and for recommending the revised paper for publication.

Reviewer #3 (Remarks to the Author):

I am mostly satisfied with the authors response to my questions and referee#1's questions. But I take issue on one point of their response, and I think this is a critical point, should be addressed more carefully before the manuscript can be accepted.

This is my original point (2), and also related to referee#1's first point: about the coupling between QD and AZ molecule wave function, especially the difference between the CR1 and CR2 transitions. The authors supposedly calculated the after transition S0 state and T1 state of the AZ molecule, and show that their electron charge densities are almost identical for these two states (Fig. S7). Frankly, I do not believe in this result. For one thing: S0 and T1 are consisted with different AZ molecular orbitals, one with AZ HOMO, one with AZ LUMO. How can they have exact same electron charge density? I feel there must be some mistakes. Note, these should not be the singlet and triplet QD exciton state before transition. Their electron charge densities can be the same if one only takes one electron QD state to construct the exciton state. But for AZ molecule, the S0 and T1 should have different single particle configurations, thus their electron charge density cannot be almost identical. Besides, it actually might be interesting just to plot the AZ HOMO and LUMO levels,

do some simple dot-product with the QD electron wave function, to make sure their coupling are similar.

Response: We thank the reviewer very much for his/her critical insights. Nevertheless, our calculation indeed showed that the electron density distributions associated with the S_0 and T_1 states of the surface-adsorbed AZ molecules (not those of QDs) were almost the same. We think this is related to the distinct nature of QD surface-adsorbed AZ molecules. As we emphasized in our paper, and also reported many prior studies, the electronic structure of surface-adsorbed AZ molecules is radically different from that of free ones. The plots in Supplementary Fig. 7 indicate that the majority of the electron densities for both S_0 and T_1 states are actually located in the vicinity of the chelated Cd atoms.

Revision: On page 10, we add the following contents (highlighted in the text):

“and both are mostly located in the vicinity of the chelated Cd atoms, a manifestation of the distinct electronic structure of QD-surface-adsorbed AZs.”

Reviewer #4 (Remarks to the Author):

I enjoyed to read the manuscript “Marcus Inverted Region of Charge Transfer from Low-Dimensional Semiconductor Materials” by Wang et al. The authors report CT in the inverted region between molecule-QD interfaces. I find the work interesting and the topic is timely as only recently Marcus Inverted region has been also found in solid-state devices (Nat. Nanotechnol. 13, 322–329 (2018) or Nat. Commun. 10, 2089 (2019)), in proton coupled charge transport (their ref 19), and now also in molecule—QD systems. At the very end of their paper the authors conclude that CT in the inverted Marcus region could be useful to improve the performance of devices. This point has been shown by Han et al. in Adv. Sci. 2021, 2100055 where the authors showed that the performance of a molecular diode was improved (30x) operating in the inverted Marcus region.

I have seen the paper and the, in general, positive review reports: I also find the paper original and important, and support publication in Nat. Commun., as the other 3 reviewers also did. In my opinion, the authors have addressed all reviewers concerns adequately. In response to Reviewer 1, the authors include now detailed TD-DFT calculations. Although the QS is only represented by two Cd ions, the calc show, nevertheless, that the electron density distributions of S_0 and T_1 are very similar.

I do agree with the authors that is very challenging to understand the atomistic structure of molecule-QD interfaces (as the state in response to comment 2 by reviewer 1), I do believe the model structures can be studied to investigate, for instance, the energy-level alignment of the molecule-QD interface, or what the most favorable binding geometries are, which in turn are important to know to quantify the molecule-QD coupling strength. Such a detailed study is out of scope of the current paper and that is why a simplified approach suffices at this stage.

Reviewer 3, comment 1, questions whether low energy processes can be follow at room temperature because at RT thermal energy (26 meV) more than enough to washout any low energy effects. The authors argue that systems with low energy barriers still be studied at RT provide one can measure such processes fast enough which I agree with. The authors provide now a clear discussion to explain this.

Reviewer 3, comment 2, has been also (partially) addressed in response to Reviewer 1. I agree with the authors that to calc the reorganization energy for the full system is very difficult to do and would justify a publication on its own right. Instead, they compare their obtained reorganization energies to those obtained from other planar molecules which is fine. In general, quantification of the reorganization energy with ab initio theories in solid-state systems is challenging and certainly falls out of scope of this work.

Comment 3 by the same reviewer relates to comment 2 and has been, in my opinion, also sufficiently well addressed.

To summarize, I believe the paper is now suitable for publication in its current form, and I believe it will form the base for many future studies..

Response: We thank the reviewer very much for his/her expert comments on our work and our responses to reviewers' prior comments, and for supporting publication of the paper. More importantly, the reviewer has introduced to us several very important studies also investigating the Marcus inverted regime, but in molecular devices. These studies in devices nicely complement our and others' spectroscopic studies on model systems.

Revision: In light of the importance of these molecular device studies on Marcus charge transfer, on page 3, we add the following contents and the citation of these papers (highlighted in the text):

“More recent studies even directly observed the inverted region in solid-state molecular junction or transistor devices and demonstrated the enhancement of device performance in the inverted region.¹⁹⁻²¹”